

# Cyanobacterial species richness and *Nostoc* highly correlated to seasonal N enrichment in the northern Australian savannah

Wendy Williams[1], Burkhard Büdel[2], Stephen Williams[1]

1.  The University of Queensland, Gatton 4343 Australia Email: wendy.williams@uq.edu.au
2.  Dept. of Biology, University of Kaiserslautern, Kaiserslautern Germany Email: buedel@bio.uni-kl.de

Correspondence to: wendy.williams@uq.edu.au

## Abstract

Boodjamulla National Park research station is situated in north-west Queensland in the dry savannah where the climate is dominated by summer monsoons and virtually dry winters. Cyanobacterial crusts almost entirely cover the flood plain soil surfaces in between the tussock grasses. Cyanobacteria fix dinitrogen that is liberated into the soil in both inorganic and organic N forms. Seasonality drives N-fixation and in the savannah, this has a large impact on both plant and soil function. In this research project, we examined the cyanobacterial species richness and bioavailable N spanning the seven months of a typical wet season. We hypothesised that cyanobacterial richness and bioavailable N would peak at the time of the heaviest rains and gradually decline in the latter stages of the wet season. We also anticipated that the abundance of N-fixing cyanobacteria would be correlated to N-fixation and N-enrichment of the surface soils. Over the wet season cyanobacterial richness ranged from 6-19 species. N-fixing *Scytonema* accounted on average across the season for 74% of the biocrust in varying proportions throughout the season. Cyanobacterial richness was highly correlated with N-fixation and bioavailable N in 0-1 cm. It was established key N-fixing species such as *Nostoc*, *Symploca* and *Gloeocapsa* significantly enriched soil N although *Nostoc* was the most influential. Total seasonal N fixation by cyanobacteria demonstrated the variability in productivity according to the number of wet days as well as the follow-on days where the soil retained adequate moisture. Based on total active days per month we estimated that N-soil enrichment via cyanobacteria would be ~ 5.2 kg ha$^{-1}$ annually which is comparable to global averages. This is a substantial contribution to the nutrient deficient savannah soils that are almost entirely reliant on the wet season for microbial turnover of organic matter. This seasonal pattern in atmospheric N-fixation and transformation to a bioavailable form was also present in C-fixation results from parallel research. Such well-defined seasonal trends and synchronisation in cyanobacterial species richness, N-fixation, bioavailable N and C fixation provide significant contributions to multi-functional microprocesses and soil fertility.



**1.0 Introduction**

The northern Australia savannahs is one of the largest natural savannahs remaining on Earth with grasslands and shrublands that cover more than 1.5 million km$^2$ (Nix et al., 2013). Over the past century there have been several major degradation

episodes, leaving about half of these important ecosystems in a degenerated state (Smith et al., 2007). It is a harsh environment, where climate shapes the ecology, distribution and abundance of resources affecting plant and animal species (Nix et al., 2013). There is a pronounced dry season, often lasting around six months, followed by violent storms and flooding rains. Across the savannah landscapes broad scale livestock grazing is the primary land use however, managing these extensive perennial grasslands and woodlands demands an approach at several different scales. On a continental scale, empirical evidence clearly

demonstrates the negative impact grazing has exerted on ecosystem structure, including key aspects of soil function (Eldridge et al., 2016). Thus, to understand the scope and variability of the northern Australian savannah, soil function is important in the context of a holistic approach to land management (Vanderduys et al., 2012) and more importantly the conservation soil microprocesses.

Once the soil surface is disturbed in any way, wind and rain erosion are significant factors that result in a loss of resources such as essential carbon (C) and nitrogen (N) stocks (Eldridge et al., 2016). Soil surface microbial communities are particularly vulnerable to disturbance by livestock, the loss of topsoil during drought, and the loss of microbial diversity especially cyanobacteria (Eldridge et al., 2010; Williams et al., 2008; Williams and Eldridge, 2011). Cyanobacterial crust communities exist where there are only small fractions of organic nutrients, where diazotrophs (bacteria that fix dinitrogen into an more

useable form), fuel soil food webs through photosynthesis and N-fixation (Elbert et al., 2012). In the northern Australian savannah the soil surfaces in between grass plants, form an almost continuous cover dominated by cyanobacteria and liverworts, occasionally lichens, bacteria, algae and fungi (Williams et al., 2014).

In these savannah landscapes, cyanobacteria seasonally reestablish and facilitate soil surface stabilisation. As phototrophic

organisms, cyanobacteria are, in mass, valuable as ecosystem engineers facilitating soil fertility on several levels (Jones et al., 1994; Eldridge et al., 2010). Newly developed cyanobacterial colonies exude slimy extracellular polysaccharides (EPS) that form organic bridges tightly binding soil aggregates and particles (Rossi et al. 2017 in press). Cyanobacterial EPS also forms a cohesive and protective layer at the soil surface, minimising the effects of wind erosion (Eldridge and Leys, 2003). EPS provides cyanobacteria the capacity to maintain fitness and sustain growth of other cohabiting species (Rossi and De Philippis,

2015; Rossi et al. 2017 in press). Cyanobacterial diversity characteristically provides a range of biochemical and physical attributes that promote resilience to microhabitat variability and climatic extremes (Rossi and De Philippis, 2015). As the biophysical structural form of the community develops, diversity of macro- and micro-organisms increase (Büdel et al., 2009).



The multi-functional species rich microbial community varies in its impact on ecosystem function, particularly nutrient cycling (Maestre et al., 2012).

Cyanobacterial mediated N results in N liberated into the soil in both inorganic and organic N forms, that in turn leads to
elevated soil inorganic N pools in the soil surfaces. (Barger et al., 2016). Bioavailable N fixed as atmospheric $N_2$ by cyanobacteria delivers a direct source for plant uptake (Mayland and Mcintosh, 1966; Belnap, 2003). This reinforces the value of the relationship between plants and cyanobacterial crusts in arid, semi-arid and savannah landscapes. In N-depleted environments nostocalean cyanobacteria develop specialised thick-walled heterocyte cells as dedicated N-fixing sites, to exclude oxygen that inhibits the N-fixing enzyme (Helm and Potts, 2012). These biophysical traits for N-uptake are critical
catalysts for cyanobacterial productivity and growth. To initiate rapid growth when conditions are favourable, storage of cyanophycin (N-storage granules) and carbohydrates are essential (Schneegurt et al., 1994). The relative abundance of cyanophycin within the cells and in the storage of polysaccharides, proteins, cell remnants and secondary metabolites in the extra-cellular matrix (ECM) provides cyanobacteria the capacity to withstand natural environmental stresses (Helm and Potts, 2012; Whitton and Potts, 2012). Communication between the cells and the environment occurs within the EPS (Rossi et al.
2017 in press). With an increase in humidity the EPS alters its rheological properties and becomes hydrophilic permitting water absorption (Helm and Potts, 2012). When it rains up to 70% of stored N is flushed out of the cyanobacterial outer matrix into to the surrounding substrate (Elbert et al., 2012; Magee and Burris, 1954; Rascher et al., 2003), where the release of N can increase if conditions are sub-optimal following desiccation (Jeanfils and Tack, 1992), thus increasing soil inorganic N pools in the upper few millimetres of the soil. (Barger et al., 2016).

The primary focus of this research has been to better understand the contribution and function of cyanobacteria on a seasonal basis to the soil ecosystem. In the northern Queensland savannah, we had previously established that cyanobacteria detect the onset of the wet season, rehydrating and resurrecting cellular functions within 24 hours of the first rains (Williams et al., 2014). Following several months with no rain throughout the dry season was a typical lead in to the summer wet season. This provided
the back drop to our research that included measuring carbon and nitrogen cycling on a seasonal basis. It was apparent from the earlier studies that even when artificially rehydrated over several days cyanobacteria would not reactivate during the dry season (Williams et al., 2014). Yet, following the first rains the cyanobacterial crust system appeared to disintegrate and regrow. It has been shown that there is a strong effect of precipitation variability on N cycling within the crust (Aranibar et al., 2004). Thus, the potential for pulses of bioavailable N over the course of the wet season was thought to be most likely connected
to rainfall events. The observed rapid growth of the cyanobacterial crusts at the height of the wet season followed by hyperproduction of EPS (Williams et al., 2014) led us to believe that bioavailable N would peak during this time.

Based on the premise that community species richness directly impacts nutrient cycling (Maestre et al., 2012) and EPS secretions maintain fitness and store crucial resources (Helm and Potts, 2012; Rossi and De Philippis, 2015), we examined the





cyanobacterial species richness and bioavailable N spanning the seven months of a typical wet season. We hypothesised that cyanobacterial richness and bioavailable N would peak at the time of the heaviest rains and gradually decline in the latter stages of the wet season. We also anticipated that the abundance of N-fixing cyanobacteria would be correlated to N-fixation and N-enrichment of the surface soils.

## 5 2.0 Methods

### 2.1 Site description

Boodjamulla National Park research station is situated in north-west Queensland in the dry tropics where the climate is heavily influenced by the summer monsoon season and described in detail in Williams et al. (2014). The lead in six months to the 2009–2010 wet season at the research site was completely dry. In late November as the humidity increased the early rainfall
was typically a couple of small rain events (0.2, 6.8 and 0.2 mm) followed by heavy rains throughout Jan (279 mm) and Feb to Apr (555 mm) (Fig 1).

### 2.2 Field sampling

To determine seasonal patterns of cyanobacterial bioavailable N, multiple sample sets were taken from November 2009 (pre-wet season rains) to May 2010 (end of wet season). Sampling was timed before, during and after major rain events to provide
a snapshot in time (also see Williams and Eldridge, 2011), and incorporated cyanobacterial surface crusts (0–1 cm) and immediately below the crust (1–3 cm) (n=5 each depth). Later in the laboratory we divided the samples into nine separate time frames that incorporated two sample sets each from early and late January and February. Each time (monthly and bi-monthly) represented at least two separate sample periods, before and after rain. For data analysis, nine-time periods were used, total n = 125 each depth. For biomass, rates of N-fixation, identification (for species richness and abundance studies), an additional
four petri dishes of 0–1 cm were collected at the same time (total n=100).

### 2.3 Laboratory analysis

### 2.3.1 Seasonal trends in N

Cyanobacterial crusts (0–1 cm) and sub-surface soils (1–3 cm) were analysed for bioavailable N ($NH_4^+ + NO_3^-$) according to Method 4, (Gianello and Bremner, 1986) and Williams and Eldridge, (2011). Seasonal N-fixation was determined through
acetylene reduction based on Hawkes and other methods for acetylene reduction assays (ARA) (Stewart et al., 1968). To complete the monthly estimates, rates of N-fixation delta $^{15}$N was calculated for each batch, and used as a conversion factor for each sample set. Petri-dish samples of cyanobacterial crusts for each month (Nov–May) were reactivated in the glasshouse for approximately two weeks. This was carried out by daily wetting to field capacity but not saturated then allowing to dry naturally. An effort was made to ensure the surface crust was unbroken. Following reactivation, 18 mm diameter plugs (six




reps per month, n=36) representative of 10% airspace were carefully removed and inserted into 40 ml glass vials with two-way septa lids. Dry weight was calculated prior to a light rewetting (~1 ml liquid) with care not to oversaturate. They were placed in natural light conditions in the glasshouse for a further two days to acclimatise prior to AR analysis. The incubation was carried out from time zero (T0) and measured at 24 hours (T2) and 48 hours (T3). In between measurements the samples

were maintained in the glasshouse at 28°C (previously determined as an optimum temperature for these crusts) and natural light conditions.

Calculation of ethylene production ($C_2H_4$ µm/mL) was carried out using the standard formula (see Hawkes, 2003; Weaver and Danso, 1994):

10          $Vhs = Vt - Vw - Vs$

where V= volume, Vhs = head space (volume of air in vial), Vt = tube (volume of tube), Vw = volume of water (ml added to sample), Vs = volume of solids (Vs = weight of sample/soil bulk density for these soils of 1.6). Daily rates were calculated by T3-T2 (48 hours – 24 hours) then converted to grams per square metre; monthly averages for $^{15}N$ were then applied as conversion values. To estimate seasonal N-fixation the mean values of N-fixation were calculated for each month and

multiplied by active days. The number of active days was based on the number of rain days and soil moisture availability measurements for key months using moisture meter data from the site, an example shown in Figure 2.

2.3.2 Carbon, Nitrogen and Biomass

Total C and total N, C:N ratio and $\delta^{15}N$ and $^{13}C$ were determined with high temperature digestion using a vario MACRO

Elemental Analyser (Elementar) and Mass Spec: Sercon Hydra 20-22 (Griffith University laboratories). For each month for ARA analysis the samples were amalgamated, dried and sieved to provide three samples for each time-period. Data was averaged to provide the conversion factors used in rates of N-fixation however there was only one replicate available for Nov–Dec as there was insufficient sample. For biomass the chlorophyll *a* extractions were carried out on the cyanobacterial soil crusts (Barnes et al., 1992) and calculated with Wellburn's (1994) equations.

**2.3.3 Cyanobacterial richness and abundance**

Morphological features and measurements were carried out from wet mounts prepared from each sample set for nine-time periods (total n=625). Abundances were determined from five subsamples of the five samples for each time (n=25). The samples for nine-time periods were rehydrated for 24 hours and examined using bright-field, phase contrast and differential interference contrast illumination systems with a Jena Zeiss and an Olympus BX51 compound microscope to a maximum

magnification of ×1000. Photomicrographs were obtained using an Olympus DP12 digital microscope camera. Identification was performed to a species level (wherever possible) in the laboratory using the nearest available keys (Anagnostidis and Komarek, 2005; Komarek and Anagnostidis, 1999).





### 2.3.4 Statistics

We examined relationships between richness of known N-fixing and non-N-fixing cyanobacteria, and N fixation using simple linear regression.

### 3.0 Results

In total three species of the nostocalean N-fixing *Scytonema* accounted on average across the season for 74% of the biocrust in varying proportions (range 55–93%) throughout the season (Table 1). Microscopic examination showed *Scytonema* was also the dominant structural component of the biocrust and this cyanobacterium was found to be the major contributor to the breakdown of the crust and its reestablishment. This took place through the disintegration of EPS and sheath material (Nov–Jan), resurrection of a portion of desiccated filaments, followed by mass release of hormogonia (asexual reproductive cells)

across Jan–Feb, then vigorous growth of new material (Figures 3-4). Average cyanobacterial biomass (Chlorophyll *a*) increased from $112.1 \pm 21.3$SE µg C$a$ g$^{-1}$ soil (Nov) throughout the wet season; peaked in Feb ($171.9 \pm 2.4$SE µg C$a$ g$^{-1}$ soil) and declined towards the end of the wet season ($153.8 \pm 19.9$SE µg C$a$ g$^{-1}$ soil) (Table 2).

### 3.1 Seasonal trends in bioavailable N

Bioavailable N was elevated in November 2009 (~6 mg NH$_4^+$ kg$^{-1}$ soil), before more than halving across January to February

followed by an exponential increase between March and May, when it peaked at >13 mg NH$_4^+$ kg$^{-1}$ soil (Fig 5). There were significant differences in depth (with more in the 0-1 cm layer) and times (except in Nov) with no significant interaction as the effect of depth was consistent across all times.

### 3.2 N productivity driven by cyanobacterial richness

Between November 2009 and May 2010 cyanobacterial richness ranged from 6 to 19 species, seven of which were known N-fixers (Table 1). Four key N-fixing cyanobacteria (*Nostoc commune, Nostoc* sp. 2, *Symploca* and *Gloeocapsa*) were significantly correlated with bioavailable N where BioN = 1.616 + 0.2072 Nrich4 (p=0.001). Of these four cyanobacteria *Nostoc* was the most influential where BioN = 0.89 + 0.475 *Nostoc* (p= 0.004). There was a strong positive relationship between bioavailable N in the top cm and total cyanobacterial richness ($F_{1,7} = 39.3$, $P < 0.001$, $R^2 = 0.83$) (Fig. 6). There was no

relationship for bioavailable N deeper in the profile (1-3 cm; $P = 0.38$). For both N fixers and non-N fixers, increasing richness was associated with an increase in N fixation at a consistent rate (Fig. 7).



### 3.3 N-fixation and chemistry

Rates of N-fixation and total season N-fixation are reported in Figures 6 and 7. Cyanobacterial richness for all species was significantly correlated to the rates of N-fixation over seven months ($P=0.002$). Average $^{15}N$ isotope across the season was 0.9 (range -0.2 to 2.1) and $^{13}C$ isotope was -19.1 to -22.0 with C:N ratios reasonably stable (average 19.1). Total C and N both

doubled across the course of the wet season (Table 2).

### 4.0 Discussion

In this study, isotopic signatures for $^{15}N_2$ across seven months of active N-fixation clearly demonstrated cyanobacteria were the primary source of bioavailable N. We showed that species richness was highly correlated with bioavailable N (Fig 6). This was further underpinned by the analysis of cyanobacterial richness that established key N-fixing species such as *Nostoc*

*commune*, *Symploca* and *Gloeocapsa* significantly enriched soil N. We had hypothesised that cyanobacterial richness and bioavailable N would pulse at times of high rainfall and gradually decline in the latter stages of the wet season as the rainfall events decreased. Even though on a seasonal basis N-fixation and N-fixed peaked at the height of the wet season, bioavailable N pulsed at the beginning of the wet season after the first rainfall, then declined before an exponential increase at the end of the wet season (Fig 5).

Dark cyanobacterial crusts dominated by species such as *Nostoc*, *Scytonema* and *Microcoleus* that were all influential in the northern savannah crusts are known for their association with high rates of N-fixation and absorption (Barger et al., 2016). At the commencement of the wet season *Scytonema*, due to its macroscopic size and colonial form was dominant (Table 1). After the first rains in November the crust structure broke down (Williams et al., 2014). Subsequently, bioavailable N was elevated

in November, most likely due to the disintegration of the EPS and some cell lysis (Williams et al. 2014), as EPS is known to store N (Otero and Vincenzini, 2003). This was followed by a reduction in bioavailable N after rain in December. We suggest this reflected the favoured investment in C-fixation by cyanobacteria (Helm and Potts, 2012) to rebuild their colonies. This was demonstrated later, where in December 2010 to January 2011 there was a net loss in productivity coinciding with rainfall and growth (Büdel et al. this journal). On the other hand, the significant increase in bioavailable N in May appeared to be

related to late season rains (in April) indicative of the investment in the storage of N in cyanophycin (granules) and EPS. Other records of seasonal influence on both C and N-fixation have been previously demonstrated however the synchrony between these events on a monthly and bi-monthly basis shows how well balanced the cyanobacterial biophysical and chemical functions were dictated by rainfall and consequently soil moisture (Büdel et al., 2009; Castillo-Monroy et al., 2010).

We had anticipated that the abundance of N-fixing cyanobacteria would be correlated to N-fixation and bioavailable N-enrichment of the surface soils. This prediction was true with a significant relationship to both N-fixation and bioavailable N at 0–1 cm depth but not for 1-3 cm depth (Fig 7). This observation points to the importance of biocrusts in the maintenance of



N in the soil thus giving access to this nutrient for the microbial communities in the crust. It had been previously shown that very high activity of carbohydrate enzymes in the biocrust hydrolysed enzymes relating to the underlying soil (Chen et al., 2014). Yet, in this study it could not be determined why there was also a significant relationship between non-N fixing species and N-fixation. There are however several explanations from other research. For example, the mutualistic beneficial relationship with heterotrophic bacteria that fix N that is subsequently taken up by the non-N fixing cyanobacterium *Microcoleus vaginatus* (Baran et al., 2015), and the role of bacteria and mycorrhizal fungi in rapid N transformation (Hawkes, 2003). In this study three species of *Microcoleus* were identified first in December but were more prominent between Feb-May (Table 2), that could provide insight into the relationships with N-enrichment and non-N-fixing cyanobacteria. It is now understood that there is a broad range of N-rich metabolites that are continually released and reabsorbed by *Microcoleus* (Baran et al., 2015). It has also been demonstrated that N-enrichment was associated with *Gloeocapsa* (Wyatt and Silvey, 1969)*, Porphyrosiphon* (Tiwari et al., 2000) and *Schizothrix* (Berrendero et al., 2016). Indeed, many cyanobacteria obtain N by scavenging from mutually shared EPS (Rossi et al. 2017 in press), or have multiple mechanisms for N-fixation either in the dark (Lüttge, 1997), through $O_2$ inhibition (Stal, 1995), and in anaerobic circumstances such as Mars (Murukesan et al., 2016), or aquatic cyanobacterial mats when submerged under water (Berrendero et al., 2016; Stewart, 1980).

## 4.1 Seasonal trends in N fixation

Total seasonal N-fixation by cyanobacteria demonstrated the variability in productivity according to the number of wet days as well as the follow-on days the soil retained adequate moisture (Fig 2) for the continuation of photosynthesis and N-fixation (Williams et al 2014). Based on total active days per month we estimated that N-soil enrichment via cyanobacteria would be ~ 5.2 kg ha$^{-1}$ seasonally. This is a substantial contribution to the nutrient deficient savannah soils that are almost entirely reliant on the wet season for microbial turnover of organic matter (Holt and Coventry, 1990). These estimations are comparable to global averages of N-fixation of 6 kg N ha-1 year-1 (Elbert et al. 2012). There are numerous examples with a broad range of values such as those of cyanobacterial crusts in grasslands from the Loess Plateau in China of 4 kg ha$^{-1}$ year$^{-1}$ (Zhao et al., 2014), or in situ results from the Negev of 10-41 kg ha$^{-1}$ year$^{-1}$ (Russow et al., 2005). Yet, many studies do not take into account a range of mitigating factors or failed to determine the $^{15}N_2$ conversion factor (Aranibar et al., 2004; Barger et al., 2016). The $^{15}N_2$ values in this study ranged between 0.3 and 2.1 (Table 2), clearly demonstrating the source of dinitrogen was cyanobacteria, whereas theoretical $^{15}N_2$ conversion rates of 3–4 or higher may be overestimating N-production (Barger et al., 2016). Isotopic measurements were taken from the cyanobacterial crusts used in AR. The conversion rate often created uncertainty although these values are comparable to other studies that have tested for $^{15}N_2$ (e.g. Aranibar et al., 2004; Russow et al., 2005). The limitations of N-fixation estimates lie in the variability of cyanobacterial cover, species richness and in this study conditions conducive to *Nostoc commune* productivity and growth.

At the height of the wet season following supersaturation of the soil profile there were two EPS hyperproduction events attributed to *Nostoc commune* (Williams et al., 2014). There is a tight linkage between rainfall, soil moisture, bioavailable N,





EPS excretion that in turn triggers a range of metabolic processes (Chen et al., 2014; Rossi and De Philippis, 2015). C and N fixation in cyanobacteria are closely interconnected as N-fixation is energy demanding and dependant on carbohydrates provided by photosynthesis (Murukesan et al., 2016). It has been reported that diazotrophic growth by cyanobacteria occurs when the N to C balance is 1 to 1.5 and the EPS is used as a sink for excess C when the C:N ratio is unbalanced (Otero and

Vincenzini, 2004), C assimilation and diversion to EPS is favoured over N fixation (Murukesan et al., 2016; De Philippis et al., 1996; Rossi and De Philippis, 2015). We were unable to make a direct comparison between C:N ratios (see Table 2) with the EPS hyperproduction. Nevertheless, some *in situ* measurements of C-fixation at this time (unpublished data) and the following year, showed that during storm events optimal temperatures, humidity, moisture and light intensity resulted in extremely high $CO_2$ uptake (Büdel et al. in this journal). With wet season storms, this would potentially result in a high C

concentration when N could prove a limiting factor. In other research authors have reported laboratory and field conditions where optimum conditions lead to EPS hyperproduction (Helm and Potts, 2012; Otero and Vincenzini, 2003; Rossi and De Philippis, 2015). This balancing mechanism (Otero and Vincenzini, 2004) could explain the decline in bioavailable N in Jan–Feb 2010 at a time when it would be anticipated that a substantial increase in N would occur. In this study *Nostoc commune*, known for its secretion of large amounts of EPS in optimum conditions, was the key species influencing N-enrichment, which

suggests that *Nostoc* growth and EPS production is an important sequence in the seasonal trends in N bioavailability. The role of EPS is to create a microenvironment for the cyanobacterial community that has low oxygen concentrations for carrying out $N_2$ fixation under anaerobic conditions (Rossi and De Philippis, 2015).

## 5.0 Conclusions

This seasonal pattern in atmospheric N-fixation and transformation to a bioavailable form was also present in C-fixation results

from parallel research for cyanobacterial crusts at the same research site (Büdel et al. in this journal). Both studies clearly demonstrate that such well-defined seasonal trends and synchronisation in cyanobacterial species richness, N-fixation, bioavailable N and C fixation provide significant contributions to multi-functional microprocesses and soil fertility. Considering the limited knowledge of N-enrichment by both heterocyte-forming cyanobacteria and cyanobacteria that rely on other strategies under different environmental conditions, we need to better understand their function especially in terms of

the importance of species richness.

Due to the vast quantities of cyanobacterial crusts present in these landscapes it follows that plant uptake of cyanobacterial mediated N is a critically important aspect of the northern Australian savannah landscape function. Land management based purely on rain-use efficiencies does not necessarily provide expected outcomes. Rain-use-efficiency is tightly coupled with

microbial activity and in this study specifically cyanobacteria provide bioavailable nutrients that would promote plant growth.



**Data and sample availability**

Research data and primary sample material (duplicates where available) for this project are filed with The University of Queensland's School of Agriculture and Food Science according to the University's policy for the use of other researchers and interested persons for future research.

**Acknowledgements**

We acknowledge the Waayni people, traditional owners of Boodjamulla National Park and thank the staff of Boodjamulla and Adels Grove. Special thanks to Tres McKenzie for onsite assistance, Ken Goulter and Katherine Raymont for lab analysis and David Eldridge for statistical advice. We also thank AgForce Qld and Century Mine for their financial and in-kind support.

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





**Table 1:** Cyanobacterial species richness and abundance over seven months (nine-time periods) expressed as a percentage. N-fixing species (shaded) were shown only if they produced heterocytes although we also determined (see main text) several non-fixing species had been associated with N-fixation.

| Cyanobacteria | NOV | DEC | JAN early | JAN late | FEB early | FEB late | MAR | APR | MAY |
|---|---|---|---|---|---|---|---|---|---|
| *Scytonema* sp. 1 | 50.6 | 93 | 78.4 | 84.4 | 64 | 52.8 | 46.2 | 32.2 | 27.2 |
| *Scytonema* sp. 2 | | | 5.8 | 8.4 | 20.6 | 1.8 | 14.8 | 21 | 9.6 |
| *Scytonema* sp. 3 | | | | | | 11.2 | 5.8 | 14.6 | 18 |
| *Nostoc commune* | 12.2 | 1.4 | 5.9 | 4 | 10.2 | 10.6 | 10.4 | 11.6 | 17 |
| *Nostoc* sp. 2 | | | | | | 0.8 | | | |
| *Symploca* sp. | 13 | 1.4 | 3.9 | 0.6 | 4.6 | 0.4 | | | 3.6 |
| *Gloeocapsa* sp. | | | | | | | 10 | 0.8 | |
| *Symplocastrum* sp. | 6.4 | 1.2 | 2 | | | 1.2 | 0.2 | 8 | 7 |
| *Schizothrix* sp. | 4 | 0.8 | 3.7 | 0.2 | | 3.8 | 2.6 | 6 | |
| *Porphyrosiphon* sp. 1 | 13.8 | 2.2 | | 1.6 | 0.4 | 4.8 | | 1.2 | 4 |
| *Porphyrosiphon* sp. 2 | | | 0.1 | 0.2 | | | 0.6 | | 0.2 |
| *Porphyrosiphon* sp. 3 | | | | | | | | 1.4 | 1.8 |
| *Porphyrosiphon* sp. 4 | | | | | | | 1 | | |
| *Microcoleus vaginatus* | | | 0.1 | 0.6 | 0.2 | | | 0.2 | |
| *Microcoleus paludosus* | | | | | | 4.8 | 1.8 | | 1.8 |
| *Microcoleus lacustris* | | | | | | 7.8 | 4 | 0.6 | 4 |
| *Oscillatoria* sp. | | | | | | | | | 0.2 |
| *Phormidium* sp. | | | | | | | 2.6 | 2.4 | 5.2 |
| *Chroococcus* sp. | | | 0.1 | | | | | | 0.4 |
| N-fixers | 75.8 | 95.8 | 94 | 97.4 | 99.4 | 77.6 | 87.2 | 80.2 | 75.4 |
| non-N-fixers | 24.2 | 4.2 | 6 | 2.6 | 0.6 | 22.4 | 12.8 | 19.8 | 24.6 |



**Table 2:** Biomass (Chlorophyll a), total carbon (C), nitrogen (N), C:N ratio and C and N isotopes for the time period from Nov-May (2009-2010).

|  | *Ca* | **C** | **N** | **C:N** | **¹³C** | **¹⁵N** |
|---|---|---|---|---|---|---|
| **NOV** | 112.1 | 0.95 | 0.06 | 17 | -19.1 | 2.1 |
| **DEC** | 146.69 | 0.9 | 0.06 | 14.9 | -19.1 | 1.0 |
| **JAN** | 120.7 | 1.61 | 0.12 | 13.9 | -17.6 | 0.9 |
| **FEB** | 171.9 | 1.8 | 0.1 | 17.2 | -16.6 | 0.9 |
| **MAR** | 150.6 | 1.58 | 0.09 | 18.1 | -15.5 | 0.7 |
| **APR** | 159.5 | 1.05 | 0.07 | 14.1 | -18.6 | 0.3 |
| **MAY** | 153.8 | 1.07 | 0.08 | 13 | -22 | 0.7 |

25



**Legends to Figures**

**Figure 1:** Daily rainfall events for 2009-2010 wet season at Boodjamulla National Park (source: bom.gov.au)

**Figure 2:** Example of ongoing soil moisture even when there is no rain for a period of time at 1-3 cm (■) and 5-10 cm (▲) for January 2010 measured with MEA TBug probes (mea.com.au)

**Figure 3:** Seasonal cyanobacterial crust functions: (a) dry cyanobacterial crust; (b) flooded crust at the commencement of

heavy rains in January; (c) rapid regrowth with EPS hyperproduction from *Nostoc* sp. and; (d) gelatinous EPS during hyperproduction phase compared with bare area being recolonised

**Figure 4:** Micrographs of cyanobacterial growth and reproduction, scale bars 20 µm: (a) *Scytonema* sp. with desiccated cells and filaments encased in outer sheath containing a high level of pigmentation (arrows), heterocytes (circled) and heavy

cyanophycin granulation; (b) *Scytonema* sp. new growth filaments illustrating hormogonia (arrow) release; (c,d) New colonies of *Nostoc* sp. illustrating EPS capsules and surrounding EPS that delivers a microenvironment for other cyanobacteria species cohabitation; (e) mature colonies of *Nostoc* sp. with heterocytes; (f) example of distinct EPS encapsulating *Nostoc* filaments within the overall colonial structure also bound together by EPS.

**Figure 5:** Seasonal trends to bioavailable N over seven months (Nov-May, 2009-2010) representing nine-time periods (early and late Jan-Feb) for 0-1 cm and 1-3 cm depths

**Figure 6:** Relationship between bioavailable N and cyanobacterial richness over nine-time periods

**Figure 7:** N-fixation increases significantly with increases in cyanobacterial richness

**Figure 8:** Rates of N-fixation over seven months

**Figure 9:** Total cyanobacterial N-fixation estimated on a monthly basis for 2009-2010 wet season






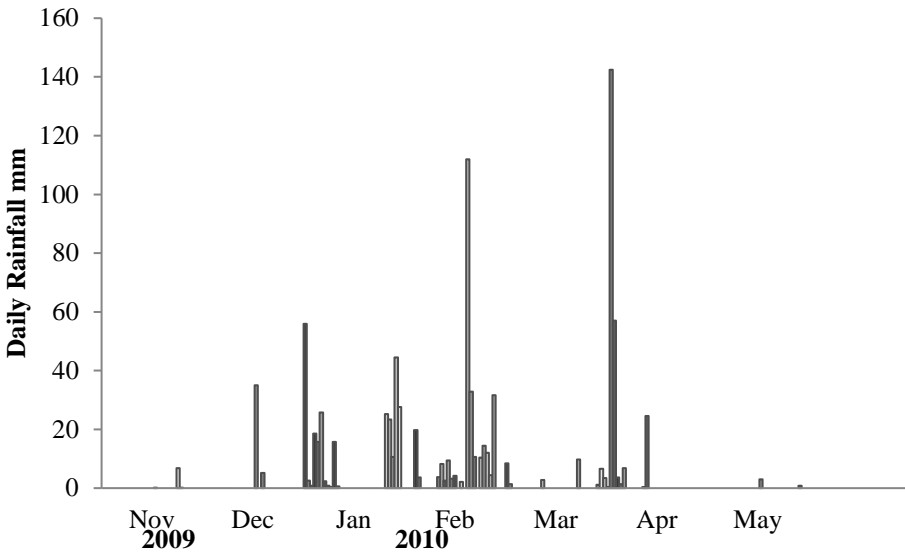

**Figure 1**





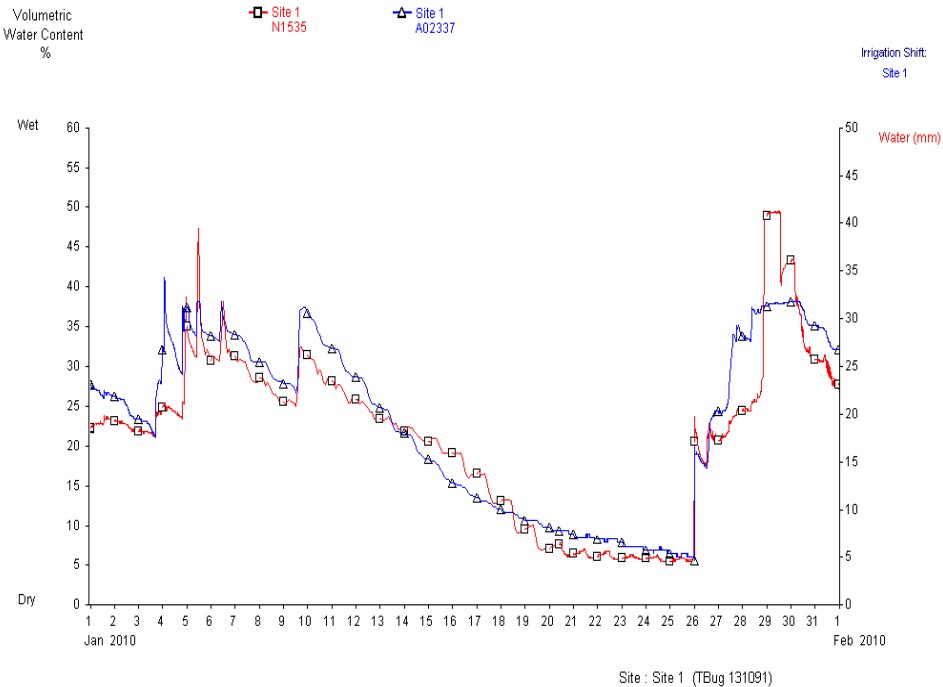

**Figure 2**





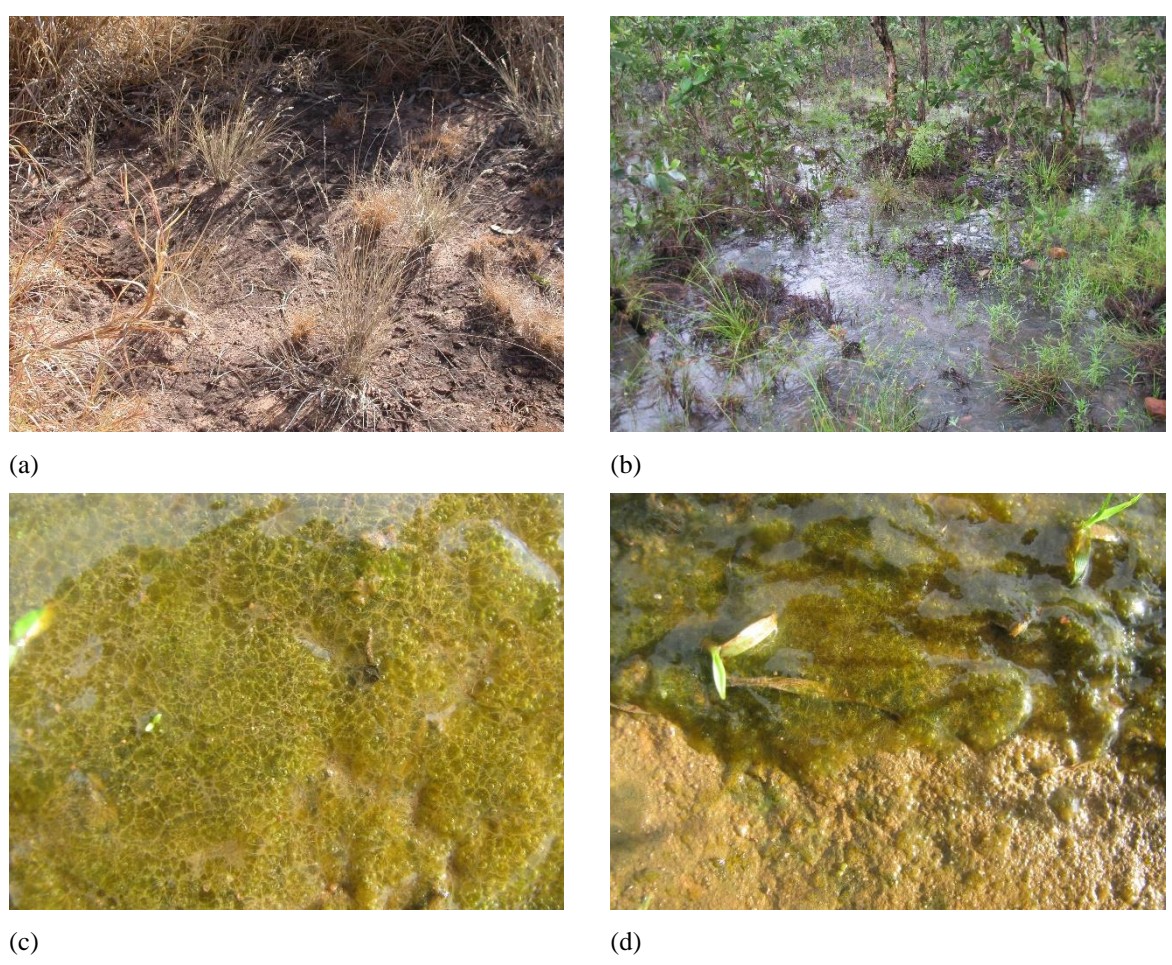

(a)

(b)

(c)

(d)

**Figure 3**



(a)

(b)

(c)

(d)

(e)

(f)

**Figure 4**





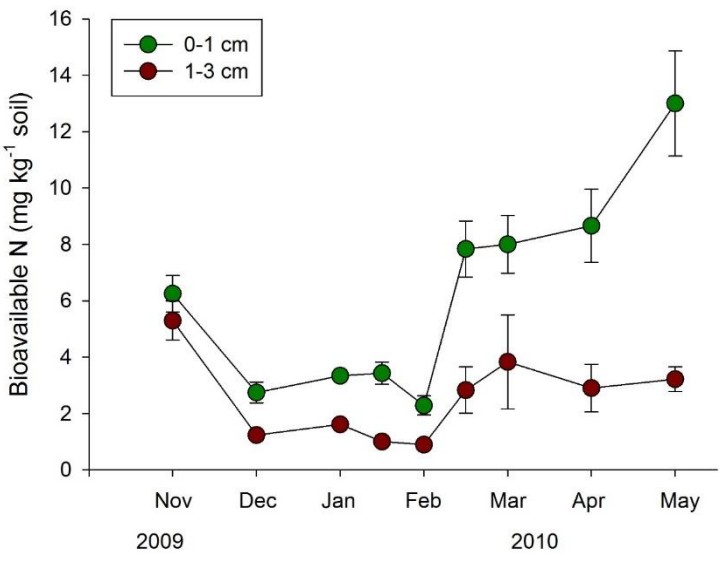

**Figure 5**

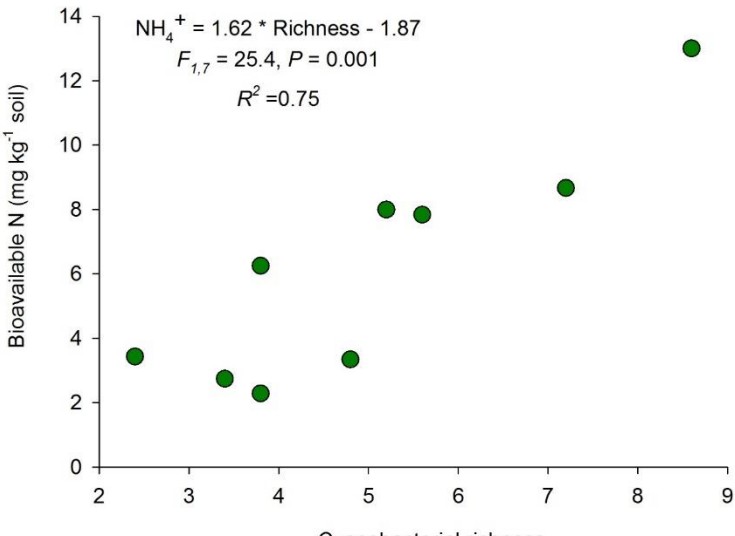

**Figure 6**

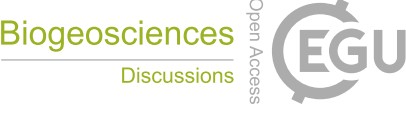



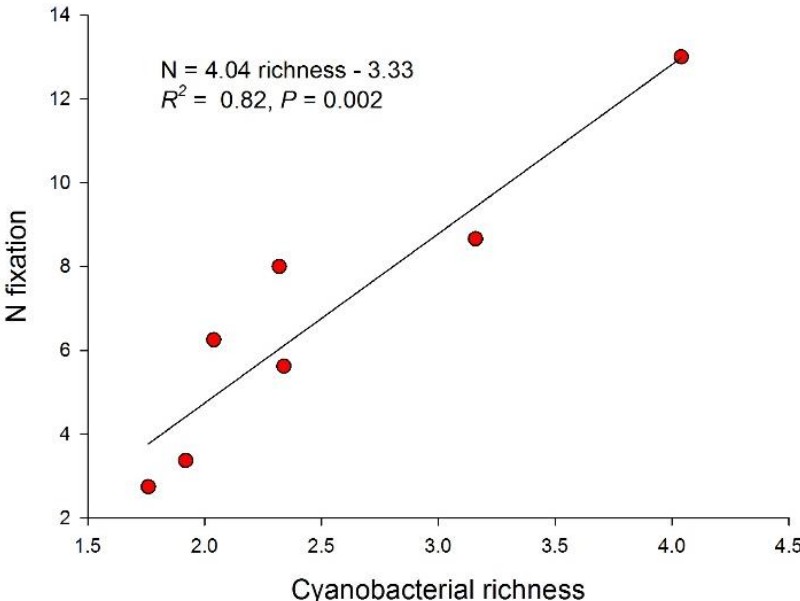

**Figure 7**





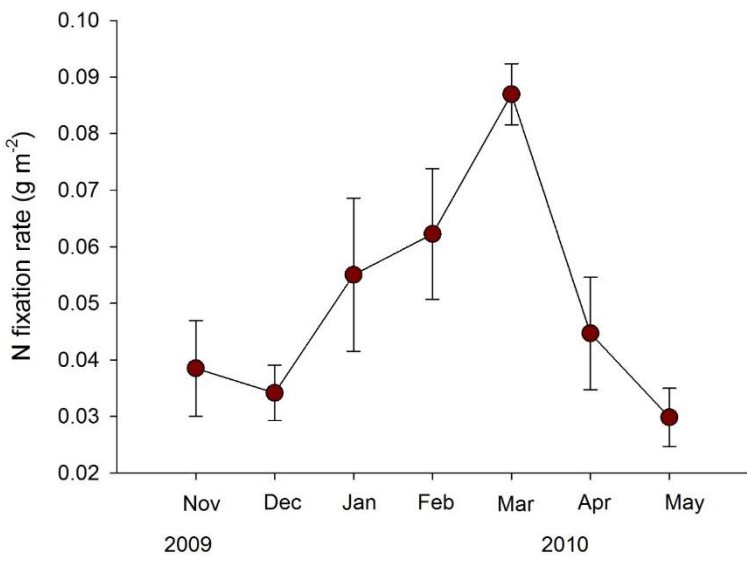

**Figure 8**



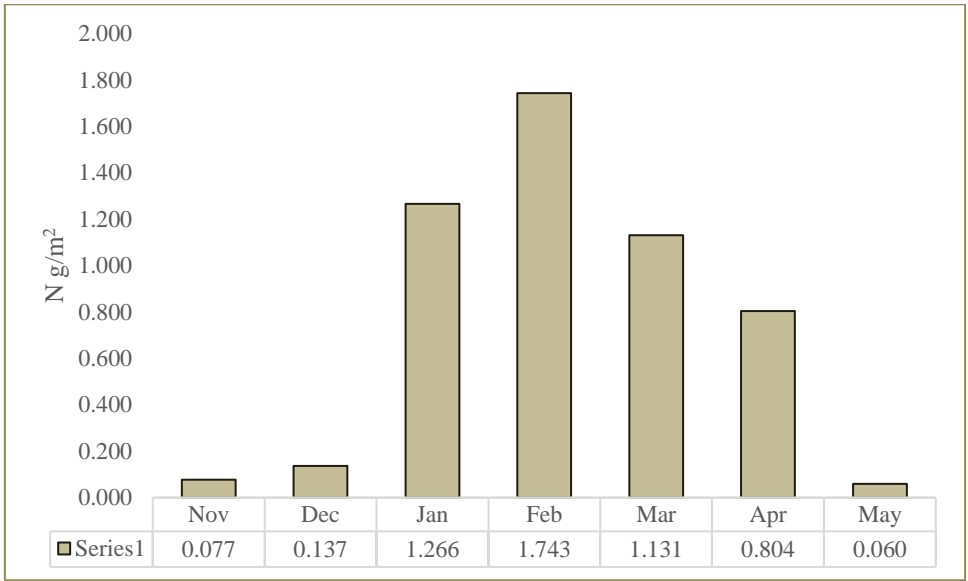

**Figure 9**

