# Peer review of "Wet-season cyanobacterial N-enrichment highly correlated with species richness and *Nostoc* in the northern Australian savannah"

_Biogeosciences, 2017_

## Referee Comment (RC1) · Anonymous Referee #1 · 5 Oct 2017

Williams et al examined Cyanobacteria species present in biocrust along with many other parameters pertaining to biological productivity (N fixation and bioavailable N) across the rainy season with the overall finding that species richness correlates with N enrichment. Overall, the findings are critical for the biocrust (and overall soil) community especially because Cyanobacteria are such important features to these ecosystems. As expected, they demonstrate evidence supporting this claim.

However, there are some flaws in the methodology that are critical to the central findings of this article. These either need to be explained more thoroughly or additional experiments conducted.

Specific areas of concern include:

-Identification of species "richness". The method by Anagnostidis is a very classical method of classifying cyanos based on morphology, but this is quite outdated and not a very reliable metric of species "richness". From this approach, how exactly is abundance and richness determined? Can you really differentiate the different species (of, for example, Microcoleus) using this visual approach? If samples are still available, the authors should consider 16S sequencing. At the very least, they should extract the pigment Scytonemin as another proxy for the abundance of N-fixers present in their sample.

-The methods overall are a bit difficult to follow. Why were samples reactivated in a glasshouse? Which samples were reactivated (just for the ARA assay)?

-Please summarize the method used to measure bioavailable N.

-How were samples collected (were they cored) and how were samples stored after collection? Were they stored at all or were they analyzed immediately?

-More details are needed for the statistics section (2.3.4) especially pertaining to section 3.2. For example, how were significant differences determined?

-In the first sentence of the discussion, it says that isotopic signatures for 15N2 clearly demonstrate that cyanobacteria were the primary source of bioavailable N. The results that are presented do not "clearly" demonstrate that. It says that nitrogen fixation is correlated with cyanobacteria richness (and with certain species), but it can not be ruled out that other N-fixers are present. This is especially true for other biocrusts where the cyanobacteria are not the primary N-fixers.

Technical corrections:

-There are many missing commas especially at the beginning of sentences throughout the entire manuscript.

[Figure]

-P1 L12: "Many cyanobacteria fix dinitrogen…" (some don't)

-P1 L18: "74% of the identified biocrust cyanobacteria in varying proportions…"

-P1 L20: This sentence doesn't make sense. Maybe deleted "It was established".

-P2 L3: "The northen Australia savannah…" (singular?)

-The Rossi et al, 2017 manuscript is now published.

-P3 L4: "Cyanobacterial mediated N fixation results in…"

-P5 L17: Bold

-P 6 L5/6: Again add "74% of the identified biocrust cyanobacteria". Also, isn't the range 51-93%? Delete "throughout the season".

-P6 L14: change to "before decreasing by more than half across January…"

-P8 L1 and 2: confusing sentence. Maybe change to "It had been previously shown that the biocrust has high carbohydrate enzymatic activity relative to the underlying soil…"

-P8 L9: change "reabsorbed" to "consumed"

-P8 L21: "global averages of biocrust N-fixation…"

---

## Referee Comment (RC2) · Anonymous Referee #2 · 11 Oct 2017

This study investigated the relationships between Cyanobacterial species richness and Nostoc highly and seasonal N enrichment in the northern Australian savannah. The results of the present study are significant to better understand their roles in influencing soil the multi-functionality. The topic is novel from my aside. Interestingly, they found that Cyanobacterial richness was highly correlated with N-fixation and bioavailable N in 0-1 cm. This paper is well organized and clearly written. However, the manuscript can be improved by shorten the abstract and conclusions for conciseness. Minor comments 2.3.4 Statistics: Please indicate what method you used for the linear regression. Line 8, P7: Here the bioavailable N consisted of $NH_4+$ an $NO_3-$, so how about dissolved organic N in the soils?

---

## Referee Comment (RC3) · Anonymous Referee #3 · 12 Oct 2017

The title of the manuscript by Williams et al. and a well-written abstract promise an interesting contribution to understanding the factors (both biotic and abiotic) that control nitrogen fixation in Australian Northern savannahs. However, while I found the topic of great interest, I also found some important weaknesses in the way the data are presented throughout the manuscript and in how the manuscript is structured. For example, in the introduction there a several cases where too general sentences leave the reader thinking what the direction of the paper will be. Unfortunately, that sensation never completely disappeared as the manuscript progressed. P3 L1-2 I don't think that this is a good citation here, as this study deals with lichens and not soil microbial communities. P3 I suggest l merging the last two paragraphs of goals and hypotheses

and rewrite them for clarity. P4 Please, report coordinates of the study site as part of the site description. P4 L19 Chlorophyll a may or may not be a good indicator of cyanobacterial biomass, but is certainly not well suited in dry environments where scytonemin is typically the most abundant pigment in soil. Therefore, I suggest avoiding the (unnecessary) use of the term biomass and just refer to pigment content, which may be a better reflection of photosynthetically active cyanobacteria. P4 L24. What is the method 4? Please, describe briefly. And the same goes for the ARA in the next line. P4 L26 and P5 L13-15. Please, explain more clearly how you estimated your conversion factor. Then I think it is probably going too far to estimate annual nitrogen fixation rates based on your ARA measurements in the lab. P6 L1-2 This is very poor description of your statistical analyses and the program used. For example, you mention a non-significant interaction in P6 L15-17 but this type of analysis is not described. P7 L3-5 Report isotopic values as delta 15N. P8 L14 Citation about nitrogen fixation in anaerobic environments in Mars does not make sense. P8 L19 5 kg of N per ha is a reasonable estimate but I would suggest stressing the need of considering this number with caution as this is only based on a few measurements in controlled conditions and I am still unconvinced about the conversion factor that you used. P9 L28-30 These conclusions come out of the blue after reading a whole paper about cyanobacterial richness, nitrogen fixation and bioavailable N. You never mentioned anything about land management and rain use efficiency before and thus I think that it would be better to restrict your conclusions to what you have really learnt with this study.

---

## Editor Comment (EC1) · E. Rodriguez-Caballero (Editor) · 20 Oct 2017

Dear Wendy William,

All three referees acknowledge the potential interest of your manuscript, but between them, they also raise a number of concerns specially regarding to the methodology used. Please carefully revise your manuscript taken into account these comments and incorporates the necessary changes.

Sincerely Emilio Rodriguez Caballero

---

## Author Comment (AC1) · 7 Dec 2017

Overall We thank all referees for providing their time and recognising the importance of this research that focuses on the critical role of cyanobacteria within biocrust communities and their contribution to soil nutrients, especially N-fixation.

We appreciate the constructive comments for improvements to the manuscript and have addressed each one below.

Referee 1 "There are some flaws in the methodology that are critical to the central findings of this article. These need to be explained more thoroughly or additional ex-

[Figure]

periments conducted." Specific areas: R1: 'Identification of species "richness". The method by Anagnostidis is a very classical method of classifying cyanos based on morphology, but this is quite outdated and not a very reliable metric of species 'richness'. From this approach, how exactly is abundance and richness determined? Can you really differentiate the different species (of, for example, Microcoleus) using this visual approach? If samples are still available, the authors should consider 16S sequencing. At the very least, they should extract the pigment Scytonemin as another proxy for the abundance of N-fixers present in their sample.'

Response: The authors acknowledge that advances in molecular techniques have shown the limitation of morphology-based metrics due to cryptic diversity, however in this case we are interested in relative differences between treatments, not absolute measures of richness. Therefore, in this application, the use of morphology-based measures (by two cyanobacterial specialists as described below) of species richness is adequate. 16S sequencing might be explored in future works to fully understand the diversity of these mat communities. Both B. Büdel (for 43 years!) and W. Williams (for 10 years) are specialists in taxonomic identification of cyanobacteria using high resolution DIC microscopy (examples of scientific publications include: Aboal et al., 2016; Anagnostidis and Komarek, 2005; Büdel et al., 2009; Komárek, 2013; Ullmann and Büdel, 2001; Williams et al., 2014; Williams and Büdel, 2012). Microcoleus paludosus, M. lacustris are first described in Australia by Williams and Büdel, (2012) together with M. vaginatus and others. We recognise that there are differences when morphological and sequenced data is studied. For example: Almost non-detectable visual differences between M. vaginatus and M. steenstruppii however there is evidence that M. steenstruppii (often described in literature as M. vaginatus) may represent several different cryptic species (Boyer et al., 2002); or in the case of Nostoc commune, ecomorphs such as N. flagelliforme that clearly have different morphological features, yet cannot be separated from N. commune through sequencing (Aboal et al., 2016). On the other hand, species which are not necessarily actively found in the soil but found with 16s sequencing might just be present with their DNA. In a recent publication, we

compared the use of both methods and found, expert knowledge presupposed, that reliable results can be achieved with both methods. Here is an extract of the Abstract from the manuscript submitted to Polar Biology now ready for resubmission after minor revision (Rippin et al., Algal biodiversity of Arctic and Antarctic biological soil crusts - Morphological vs. molecular approaches): Morphological identification using light microscopy and the annotation of ribosomal sequences taken from metatranscriptomes. The analyzed samples were collected at Ny-Ålesund, Svalbard, Norway, and the Juan Carlos I Antarctic Base, Livingston Island, Antarctica. This study is focused on the following taxonomic groups: Klebsormidiophyceae, Chlorophyceae, Trebouxiophyceae, Xanthophyceae and Cyanobacteria. In total, 143 and 103 genera were identified in the Arctic and Antarctic sample, respectively, via both approaches, while 15 and 7 taxa were determined concordantly. Hence, some genera were identified only by one of the two techniques. In general, the molecular analysis indicated a higher degree of microalgal and cyanobacterial diversity (about 11 times higher). In terms of eukaryotic algae, the two sampling sites displayed comparable genus counts while the cyanobacterial diversity was much higher in the BSC from Ny-Ålesund. Furthermore, the Arctic and the Antarctic BSC shared a total of 63 microalgal species. For the first time, the presence of the genera Chloroidium, Ankistrodesmus and Dunaliella in Polar Regions was determined. Overall, these findings illustrate that only the combination of morphological and molecular techniques, in contrast to one single approach, reveals a higher degree of biodiversity for complex communities such as polar BSCs. The description of species richness in this manuscript has been derived from the separation of different cyanobacterial species through morphological features and identified to a species level where possible (see p5, Section 2.3.3). Please note the reference list used for identification has been updated (see p5, Section 2.3.3) to include more recent taxonomic literature. We respectfully disagree that identifying cyanobacteria morphologically is outdated, inasmuch that many species from the Australian continent are undescribed both in literature and sequences not available in current gene libraries (e.g. (Chilton et al., 2017 and Williams, Chilton and Alchin, unpublished data). Samples have been

preserved for future sequencing analysis (when funds are available) that will be part of a larger biogeographical study. If scytonemin has been used as a proxy for N-fixers it assumes that only diazotrophic cyanobacteria produce scytonemin which is not the case. The position of filaments within the mat profile (and their relative exposure to UV light) is a large determinant of scytonemin production. The authors respectfully disagree that additional data relating to scytonemin would add to the records presented here of known N-fixing species.

R1: The methods overall are a bit difficult to follow: Response – areas of concern are addressed below.

R1: 'Why were the samples reactivated in the glasshouse' and 'which samples were reactivated (just for ARA)?' Response – we have added a sentence into the methods and clarified for each section the treatment of each sample set for the different analyses. Added into methods Section 2.3.1, p5, L10 Full resurrection during the wet season (when humidity increases) was critical due to the inability of these cyanobacteria to reactivate during the dry season (see Williams et al. 2014).

R1: "Please summarise method used to measure bioavailable N

Response: Added into methods p4 Section 2.3.1 Cyanobacterial crusts (0–1 cm) and sub-surface soils (1–3 cm) were analysed for bioavailable N ($NH_4+$ + $NO_3Âŕ$) according to Method 4, (Gianello and Bremner, 1986) and Williams and Eldridge, (2011). The samples had been immediately dried in the field (>40°C) and returned to the laboratory where they were stored dry. Duplicates for each of nine-time periods and each depth (minimum four reps) were sieved (1.86 mm) and weighed (20 mg) for both hot and cold analysis (total across all time periods for each depth, 0–1 cm n=125, 0–3 cm n=99). This method determines ammonium-N ($NH_4+$) produced from organic soil N when the soil is heated with 2M KCl in a stoppered tube at 95°C for 16 hours. $NH_4+$ is determined by the difference between the $NH_4+$ liberation during the hot distillation of 20 mg soil and the $NH_4+$ present prior to heating (Gianello and Bremner, 1986) and

provides an index for bioavailable (mineralizable) N at the time of sampling (also see Tongway and Ludwig, 1996; Williams and Eldridge, 2011).

R1: "How were the samples collected (were they cored) and how were the samples stored after collection? Were they stored at all or were they analysed immediately?

Response: Added to P4, Section 2.2 L16-18,: 'Cyanobacterial surface crusts and subsoils were removed using a 10 cm metal spatula to extract exactly 0–1 cm vertical depth with care to include the complete sample, followed up by 1–3 cm depth directly below the crust.' In Section 2.3.1 we have added the dry storage and time frame for when the samples were analysed. 'The samples had been immediately dried in the field (>40°C) and returned to the laboratory where they were stored dry and analysed after the end of the wet season (2010).'

R1: "More details are needed in the statistics section (section 2.3.4) especially pertaining to section 3.2."

Response: The following detail has been added to improve the description of the statistics and provides an explanation of the analysis of the results presented in section 3.2 2.3.4 Statistics 'We used linear regression models to examine potential relationships between bioavailable N and cyanobacterial richness separately, for N-fixing and non-N-fixing cyanobacteria (see Table 1). We examined differences in bioavailable N between the two depths across time with mixed-models ANOVA. Our model had two strata, one that accounted for the differences among the nine time periods, and a second stratum accounting for depth and its interaction with time. All of these analyses were run in Minitab Version 16.1.0 (2010). Least Significant Difference (LSD) testing was used to examine differences in means among the nine time periods. Tests for homogeneity of variance, independence and normality in the data, using Levene's test and other diagnostic tools in the Minitab (2010) statistical package, indicated that no transformations were necessary.'

R1: "In the first sentence of the discussion it says that isotopic signatures for 15N2

clearly demonstrate that cyanobacteria were the primary source of bioavailable N. The results that are presented do not "clearly" demonstrate that. It says that nitrogen fixation is correlated with cyanobacterial richness (and with certain species) but it cannot be ruled out that other N-fixers are present. This is especially true for other biocrusts where cyanobacteria are not the primary N-fixers."

Response: In the first sentence of the discussion (Section 4) we have changed the start of the first paragraph to read: 'In this study, isotopic signatures for $\delta$ 15N across seven months of active N-fixation demonstrated cyanobacteria were likely to be the primary source of bioavailable N (Evans and Ehleringer, 1994), although it is now understood that microbial resource partitioning on a microscale takes place within the biocrust strata (Baran et al., 2015). Given the rapid wetting following the dry season, potential cell lysis (Williams et al., 2014; Williams and Eldridge, 2011) and the presence of other N-fixers (Hawkes, 2003), it is difficult to separate the exact source of bioavailable N (e.g. Baran et al., 2015; Dojani et al., 2011; Johnson et al., 2007).....'

R1: Technical corrections:

R1: P1, L12: Response: Added 'Many' R1: P1, L18: Response: Changed sentence to read: 'Over the wet season cyanobacterial richness ranged from 6–19 species. N-fixing Scytonema accounted for seasonal averages between 51–93% of the biocrust.' R1: P1, L20: Response: deleted 'It was established...' R1: P2, L3: Response: changed to 'the northern Australian savannah...' R1: P2: Response: Rossi et al. 2017 added as reference R1: P4, L5: Response: added the word 'fixation' R1: P5, L17: Response: Formatted heading to bold R1: P6, L5/6: Response: revised sentence to 'Three species of the nostocalean N-fixing Scytonema accounted for 74% of the biocrust in varying proportions (range 55–93%) throughout the season (Table 1).' R1: P6, L14: Response: changed to '...before changing to more than half across January to February....' R1: P8, L1/2: Response: Sentence changed to: 'It had been previously shown that the biocrust had high enzymatic relative to the underlying soil (Chen et al., 2014).' R1: P8, L9: Response: changed 'reabsorbed' to 'consumed' R1:

P8, L21: Response: added the word 'biocrust'

---

## Author Comment (AC2) · 7 Dec 2017

Overall We thank all referees for providing their time and recognising the importance of this research that focuses on the critical role of cyanobacteria within biocrust communities and their contribution to soil nutrients, especially N-fixation.

We appreciate the constructive comments for improvements to the manuscript and have addressed each one below. Referee 2 R2: The manuscript can be improved by shortening the abstract and conclusion. Response: The abstract has been reduced from 330 words to 255 words: 'Abstract Boodjamulla National Park research station is situated in north-west Queensland dry savannah where the climate is dominated

by summer monsoons and virtually dry winters. Cyanobacterial crusts almost entirely cover the flood plain soil surfaces in between the tussock grasses. Seasonality drives N-fixation and in the savannah, this has a large impact on both plant and soil function. Many cyanobacteria fix dinitrogen that is liberated into the soil in both inorganic and organic N forms. We examined cyanobacterial species richness and bioavailable N spanning seven months of a typical wet season. Over the wet season cyanobacterial richness ranged from 6–19 species. N-fixing Scytonema accounted for seasonal averages between 51–93% of the biocrust. Cyanobacterial richness was highly correlated with N-fixation and bioavailable N in 0-1 cm. Key N-fixing species such as Nostoc, Symploca and Gloeocapsa significantly enriched soil N although Nostoc was the most influential. Total seasonal N fixation by cyanobacteria demonstrated the variability in productivity according to the number of wet days as well as the follow-on days where the soil retained adequate moisture. Based on total active days per month we estimated that N-soil enrichment via cyanobacteria would be $\sim$ 5.2 kg ha-1 annually which is comparable to global averages. This is a substantial contribution to the nutrient deficient savannah soils that are almost entirely reliant on the wet season for microbial turnover of organic matter. Such well-defined seasonal trends and synchronisation in cyanobacterial species richness, N-fixation, bioavailable N and C fixation (this journal) provide important contributions to multi-functional microprocesses and soil fertility.' The second paragraph from the conclusion has been removed (also see comments from R3)

R2: "Minor comments 2.3.4 Statistics: Please indicate what method you used for linear regression." Response: Statistics section now has this information included: 'We used linear regression models to examine potential relationships between bioavailable N and cyanobacterial richness separately, for N-fixing and non-N-fixing cyanobacteria (see Table 1). We examined differences in bioavailable N between the two depths across time with mixed-models ANOVA. Our model had two strata, one that accounted for the differences among the nine time periods, and a second stratum accounting for depth and its interaction with time. All of these analyses were run in Minitab Version 16.1.0

(2010). Least Significant Difference (LSD) testing was used to examine differences in means among the nine time periods. Tests for homogeneity of variance, independence and normality in the data, using Levene's test and other diagnostic tools in the Minitab (2010) statistical package, indicated that no transformations were necessary.'

R2: P7, L8: "Here the bioavailable N consisted $NH_4^+$ and $NO_3^-$, so how about dissolved organic N in soils?" Response: We did not fractionate the forms of N therefore did not measure DON however with the 2M KCL hot extraction it is possible that some DON could be converted and in this case the measurement would include that. The primary focus for this project was to understand bioavailable N that may have entered the system via cyanobacteria.

———————————————

---

## Author Comment (AC3) · 7 Dec 2017

Overall We thank all referees for providing their time and recognising the importance of this research that focuses on the critical role of cyanobacteria within biocrust communities and their contribution to soil nutrients, especially N-fixation.

We appreciate the constructive comments for improvements to the manuscript and have addressed each one below. Referee 3 R3: "….in the introduction there are some too general sentences leaving the reader what direction the paper will be. . ." Response: In the first three paragraphs of the introduction we have deleted some ambiguous and general sentences and rearranged certain sections so the concepts flow from the land-

scape to the microbe better. We appreciate the referee's comments here as it assists greatly in refining the manuscript. R3: P3, L1: "I don't think this is a good citation. . ..as it deals with lichens. . ." Response: This sentence and citation has been removed in the revision of the introduction. R3: P3: "I suggest rewriting the last two paragraphs for clarity and merging them." Response: As suggested these paragraphs have been revised with a couple of sentences removed to improve flow and merged into one. R3: P4, ". . .report coordinate of study site." Response: Coordinates have now been included. R3: P4, L19: "Chlorophyll may or may not be a good indicator of cyanobacterial biomass, but is certainly not well suited to dry environments where scytonemin is typically the most abundant pigment in soil. Therefore, I suggest avoiding the unnecessary use of the term 'biomass' and just refer to pigment content, which may be a better reflection of photosynthetically active cyanobacteria." Response: As suggested the term biomass has been used sparingly and replaced with pigment content defined as chlorophyll a (see Section 2.3.2 and others throughout). Response: Inserted into Section 2.3.1, L2: 'To estimate seasonal N-fixation the mean values of N-fixation were calculated for each month and multiplied by active days. As we have clear data that indicates the periods of activity for these biocrusts (see Williams et al., 2014 and Büdel et al. this journal) the number of active days was based on the number of rain days and soil moisture availability measurements for key months using moisture meter data from the site, an example shown in Figure 2.' R3: P4, L24: "What is method 4? Please describe briefly. And the same goes for ARA" Response: Method 4 descriptions have been addressed with the response to R1 with a more detailed description in methods Section 2.3.1 (shown in this document P3). ARA description has been improved in the second paragraph.

R3: "Please explain more clearly how you estimated your conversion factor" Response: The conversion factor was derived from isotopic analysis. We clarified the origin of the delta 15N conversion factor in the methods (P5, section 2.3.1): 'Seasonal N-fixation was determined through acetylene reduction based on Hawkes (2003) method for acetylene reduction assays (ARA) (Stewart et al., 1968). To complete the monthly

estimates of rates of N-fixation, delta 15N of the crust was calculated for each sample, and used as a conversion factor for each month's sample set (also see section 2.3.2). Petri-dish samples of cyanobacterial crusts (reserved for AR) for each month (Nov–May) were reactivated in the glasshouse for approximately two weeks. Full resurrection during the wet season (when humidity increases) was critical due to the inability of these cyanobacteria to reactivate during the dry season (see Williams et al. 2014).'

'In Section 2.3.1 additional information has been provided to clarify seasonal estimations: 'Monthly averages for $\delta$ 15N (derived from ARA samples, see Section 2.3.2) were then applied as conversion values and care was taken to ensure units were equivalent prior to final calculations. To estimate seasonal N-fixation the mean values of N-fixation were calculated for each month and multiplied by active days. As we have clear data that indicates the periods of activity for these biocrusts (see Williams et al., 2014 and Büdel et al. this journal) the number of active days was based on the number of rain days and soil moisture availability measurements for key months using moisture meter data from the site, an example shown in Figure 2.'

R3: "...I think it is going too far to estimate your N-fixation rates based on ARA rates measured in the lab..." and "P8, L19 5 kg of N per ha is a reasonable estimate but I would suggest stressing the need for considering the number with caution as this is only based on a few measurements in controlled conditions...." Response: We appreciate that it is difficult to be certain that the estimation of 5 kg N per ha is accurate given the changing nature of field conditions. The authors stand by this estimation as a reasonable attempt to calculate the contribution of cyanobacterial crusts to N for this season at this site. However, at R3's suggestion we have added a statement to reinforce the fact that this is an estimation and there are many variables that could alter this figure. Section 4.1 has an additional sentence and some edits to read: 'Based on total active days per month we estimated that N-soil enrichment via cyanobacteria would be $\sim$ 5.2 kg ha-1 seasonally. This estimation must be treated with caution as in the field there are multiple environmental variables that could result in this figure

being higher or lower. Notwithstanding, this indicates a substantial contribution to the nutrient deficient savannah soils that are almost entirely reliant on the wet season for microbial turnover of organic matter (Holt and Coventry, 1990). These estimations are comparable to global averages of biocrust N-fixation of 6 kg N ha-1 year-1 (Elbert et al. 2012).' And further down in the same paragraph we had already stated: 'Other limitations of N-fixation estimates lie in the variability of cyanobacterial cover, species richness and in this study conditions conducive to Nostoc commune productivity and growth.'

R3: P6, L1-2 "This is a very poor description of statistics.... non-significant interaction not described" Response: The statistics section has been rewritten and includes aforementioned descriptions (see R1 insertion P4, this document).

R3: "Report isotopic values as delta 15N" Response: This has been altered throughout the manuscript.

R3: "Citation about N-fixation on Mars does not make sense" Response: This citation was merely an example of anaerobic N-fixation but the words "such as Mars" have been removed so as not to detract from general point.

R3: P9, L28-30 "....these conclusions come out of the blue..." Response: In line with R2 and R3's comments this paragraph has been removed to make the conclusion more concise and to the point.

---

## Author Response (AR2)

Editors requests and responses

1) In figure 1 Please modify mm to (mm)

Response: Figure 1 has been replaced with (mm) modified.

2) In figure 2, it is not clear what N1535; A02337 TBUG 131091 means. Are they codes for the different sampling locations? It will be desirable to use a better description of what they mean.
Moreover, as I see in the figure this graph corresponds to site 1, but sites are not defined within the methods. Does Site 1 correspond to Boodjamulla National Park Research Site? or on the contrary it represent an specific sampling location.
Please clarify this also in the page 5 of the main document "The number of active days was based on the number of rain days and soil moisture availability measurements for key months using moisture meter data from the site, an example shown in Figure 2". What secondary Y axis represent

Responses: Figure 2 has been replaced with the annotations removed. These annotations were embedded in MEA (moisture probes) data logger program.

In Figure 2's description site one is verified as the Boodjamulla Park research site.

The right left and right Y axis have been clearly identified and in the new graph the rainfall events are inserted which now makes the right hand Y axis make sense!

The following changes have been made to the text on P5 "To estimate seasonal N-fixation, the mean values of N-fixation were calculated for each month and multiplied by photosynthetically active days. As we have clear data that indicates the periods of activity for these biocrusts (see Williams et al., 2014 and Büdel et al. this journal), the number of photosynthetically productive days were calculated by: the number of rain days plus the number of days that soil moisture was available following rain. This was based on moisture meter data from the site (example shown in Figure 2) and *in situ* photosynthetic yield tests previously carried out by Wiiliams et al., (2014)."

2) In Figure 4 please increase font size in the scale bar.

Response: it is not really possible to increase the font size on the scale bar as it is embedded in the microscope image however I have highlighted in the description the scale of the error bar so once the description is below the images it will be very clear as to the scale of the micrograph.

3) In Figure 5, what uncertainty bars represents? Standard error, standard deviation?

Response: SEM annotation added to Figure 5 description.

5) In Figure 6: It is possible to add regression line, in a similar way as in figure 7?
I would suggest homogenizing regressions equations in figure 6 and 7: equation; F factor; p value; r2. Moreover, units for N fixation are necessary

Response: Regression line added to Fig 7, N fixation units added, equations homogenised.

6) In Figure 8, what uncertainty bars represents? Standard error, standard deviation?

Response: SEM annotation added to Figure 8 description.

7) In Figure 9 y axis is missing; What N means (N fixation, units not in the correct format (g m -2). What series 1 means and what numbers represent? Please clarify

Response: Figure 9 replaced as numbers were repetitious of data represented on Y axis, description added and corrected.

[revised manuscript text omitted]

(a)

(b)

(c)

(d)

5 **Figure 3**

[Figure]

(a)

(b)

(c)

(d)

(e)

(f)

**Figure 4**

[Figure]

5      **Figure 5**

[Figure]

**Figure 6**

[Figure]

**Figure 7**

[Figure]

**Figure 8**

[Figure]

**Figure 9**

---

## Author Response (AR3)

**Editors requests and responses**

Figure 2: New graph do not coincide with the previous version, please can you explain this issue?

The replacement for Figure 2 (23 Jan to 23 Feb 2010) actually was a different time frame to the original Fig 1 (1 Jan to 31 Jan 2010) so if you look at the overlap period (23-31 Jan) you will see this is the same data. The reason I replaced the image is that the second one for Feb included rainfall data which was not available in this software for the Jan data.

Why there is a decrease of soil water content after rainfall events from 10/feb/2010 to 20/feb/2010?

The trend was a decrease due to the small size of the rainfall events however the image also shows a slight increase particularly in surface moisture immediately after each event. The short sharp declines in the blue line (5-10 cm) illustrates the drying out of the subsurface water as there had been no larger rain events for a while to refill the profile. I have now revised detail to Fig 2 description to read:

**Figure 2:** Example of soil moisture (volumetric water content %) patterns plotted against rainfall (mm) for 23 January to 23 Feb 2010 illustrating the retention of soil moisture after rain. A drying trend following significant rain was intermittently supplemented by smaller rain events, then after 45 mm of rain on the 20th Feb the soil profile was recharged from 15% to 50%. Moisture was measured with MEA TBug probes (mea.com.au).

Moreover, I recommend to include series name Probe depths at 1–3 cm (red squares) and 5–10 cm (blue triangles) in the graphs, instead of Site 1 and Site 1) and rainfall as title for second y axis.

These changes have been made

It will be good to use raw data to reedit this figure in a better graphical resolution

I have made necessary improvements and upgraded quality of image.

In figure 5; and 8: SEM should be explained

Completed

Figure 9: Is it possible to introduce uncertainty in monthly values. Black line representing Y axis could be good

Completed

[revised manuscript text omitted]

(a)

(b)

(c)

(d)

**Figure 3**

[Figure]

(a)

(b)

(c)

(d)

(e)

(f)

**Figure 4**

[Figure]

5        **Figure 5**

[Figure]

**Figure 6**

[Figure]

**Figure 7**

[Figure]

**Figure 8**

[Figure]

**Figure 9**

---

## Author Response (AR4)

**Editors requests and responses**

In Figure 1, what I want to understand is why soil water content in prove (5-10cm depth) shows several peaks of reduced water content just after each small rain peak occurring between 11-20 Feb (E.G. from 15 to 5% during few hours and then it rises again). Are these artifacts?

5 The rainfall events are manually entered in a 24-hour cycle (daily) and the soil moisture was measured hourly therefore the exact location on the graph (rainfall) may not be precise. As you can observe the decreases or increases are not all the same so I don't believe this is an artefact of the logger. During these time periods it can be extremely hot with very high evaporation and high use of ground moisture by shrubs, grasses and crusts. This can be followed by a storm and sometimes also heavy dewfall (2 mm) so there can also be discrepancies between soil moisture at the surface and subsurface. The

10 probes were three-pronged and the measurement was an average of those three measurements. If you would like a more detailed description in the methods or figure description please let me know. Sometimes it is difficult for a reader unfamiliar with these environments to understand what actually occurs, which is why these measurements are an important component of the explanation as to why the crusts can continue to function even with no rain or small rainfall events.

15 Figure 6: Relationship between bioavailable N and cyanobacterial richness over nine x time periods. Please clarify what x means

I have removed the anomaly of x (throughout ms) which was more a way of trying to make nine time periods read clearly however I now realise it is a bit confusing!

20 In figure 7, y axis still wrong (mg-1 day-1) please modify to (mg day-1). What about "Relationship between cyanobacterial richness and N-Fixation rates. As showed, N-fixation increases significantly with increases in cyanobacterial richness

Corrected

Caption of figure 8: Each figure has to be auto-explicative, thus I recommended replacing error bars SEM. to error bars

25 represent standard error of the mean (SEM). Idem for EPS acronym showed on captions of figure 3 and 4

Corrected

What is the different between N2 fixation rate (g m-2) showed at figure 8 and N2 fixation g m2 showed in figure 9? In figure 9, please modify units to (g m-2)

30 Figure 9 represents the total mean N-fixation multiplied by no. of active days - this adds up to a seasonal (seven month) total of ~5.2 g m$^{-2}$ which is equivalent to 5.2 kg ha$^{-1}$ the seasonal figure quoted in the manuscript. Units in Fig 9 corrected.

Figure 8. does this means that In January (as an example) total daily N fixation was 0.055 g m-2?. This is what I understand from the graph, otherwise units are missing – Yes this is the correct interpretation – I have added daily in figure description

35 to clarify.

Figure 9: does this means that In January (as an example) total N fixation was 1.25 g m-2?. This is what I understand from the graph, otherwise units are missing

Yes, this is correct as you have explained above.

[revised manuscript text omitted]

(a)

(b)

(c)

(d)

**Figure 3**

[Figure]

(a)

(b)

(c)

(d)

(e)

(f)

**Figure 4**

[Figure]

5        **Figure 5**

[Figure]

**Figure 6**

[Figure]

**Figure 7**

[Figure]

5    **Figure 8**

[Figure]

**Figure 9**

---

## Author Response (AR5)

**Editors requests and responses**

Figure 2 still need some corrections. Please check small peaks on soil water content. This means that there is a decrease of more than 20% of soil water content for about 1h after a rainfall, or during a rainfall, and then it increase again to previous values. These packs are independent of the general drying trend that it is visible form 2/Feb to 20/Feb. As you see, the effect of individual rainfalls can be seen by slight increase in soil water content of 3-4%, but these packs do not seems to be related to this effect (E.G on 16/Feb there is no rainfall and same pattern can be observed). Please can you explain this issue

Response:

I believe these are some anomalies of the sensor so I have changed Figure 2 to read:

[revised manuscript text omitted]

(a)

(b)

(c)

(d)

**Figure 3**

[Figure]

(a)

(b)

(c)

(d)

(e)

(f)

**Figure 4**

[Figure]

5          **Figure 5**

[Figure]

$NH_4^+ = 1.62 * Richness - 1.87$

$F_{1,7} = 25.4, P = 0.001$

$R^2 = 0.75$

**Figure 6**

[Figure]

**Figure 7**

[Figure]

5 **Figure 8**

[Figure]

**Figure 9**